# FMI-CAECD: Fusing Multi-Input Convolutional Features with Enhanced Channel Attention for Cardiovascular Diseases Prediction

**DOI:** 10.3390/s24227160

**Published:** 2024-11-07

**Authors:** Tao Lin, Mengyao Fan

**Affiliations:** The School of Computer Science & Information Engineering, Shanghai Institute of Technology, Shanghai 201418, China; 226141100@mail.sit.edu.cn

**Keywords:** convolutional neural network, disease prediction, feature extraction, risk assessment, Shapley additive explanations

## Abstract

Cardiovascular diseases (CVD) have become a major public health problem affecting the national economy and social development, and have become one of the major causes of death. Therefore, the prevention, control and risk assessment of CVD have been increasingly emphasized. However, current CVD prediction models face limitations in capturing complex relationships within physiological data, potentially hindering accurate risk assessment. This study addresses this gap by proposing a novel Framework for Multi-Input, One-dimensional Convolutional Neural Network (1D-CNN) with Attention Mechanism for CVD (FMI-CAECD). This framework leverages the feature extraction capabilities of Convolutional Neural Network (CNN) alongside an Attention Mechanism to adaptively identify critical features and non-linear relationships within the data. Additionally, Shapley Additive Explanations (SHAP) analysis is incorporated to provide deeper insights into individual feature importance for disease prediction. Performance evaluation on the BRFSS 2022 dataset demonstrates that FMI-CAECD achieves superior accuracy (97.45%), sensitivity (96.84%), specificity (95.07%), and F1-score (92.44%) compared to traditional machine learning baselines and other deep learning models. These findings suggest that FMI-CAECD offers a promising approach for CVD risk assessment.

## 1. Introduction

CVD constitutes a significant global public health concern, ranking as the leading cause of mortality worldwide [1]. CVD is widely recognized as a grave global health challenge, responsible for a significant number of deaths and imposing a substantial burden on healthcare systems worldwide. Various unhealthy lifestyle choices and behavioral habits are associated with an increased risk of disease incidence. The American Heart Association has identified several factors that may increase the risk of developing CVD, including sleep disturbances, advanced age, and obesity [2]. These factors may operate independently or interactively, thereby enhancing the potential for an individual to develop CVD. Therefore, the early diagnosis, effective prevention, and treatment of CVD are crucial.

Machine learning (ML), a burgeoning field within computer science, has emerged as a powerful tool in the medical domain. ML applications in tumor segmentation and disease prediction have yielded promising results. Analogous to tumor segmentation and other disease prediction applications, machine learning techniques hold promise for CVD analysis. Within the realm of ML, deep learning (DL) has garnered increasing attention for its proficiency in data processing and capacity for adaptive feature extraction from complex datasets. These characteristics position DL as a valuable tool for tasks such as cardiovascular disease analysis.

However, current CVD prediction models face many challenges when dealing with structured data. First, most existing models rely heavily on traditional machine learning methods such as logistic regression (LR), random forests (RFs), support vector machines (SVMs), and so on. These methods often lack effective strategies to capture the complex interrelationships among features when dealing with different features, such as numerical, categorical, and ordinal features. Second, these methods are sensitive to high-dimensional features and data imbalance, which may lead to poor model performance in small sample prediction or rare event detection. Some existing deep learning methods, such as single-input CNN, usually focus on a single type of data input, which may be limited in fully utilizing the interactive information of different feature types. In addition, many existing models have limited interpretability, making it difficult for clinicians to understand the basis of the model’s predictions, which affects the application of these models in actual clinical practice.

Based on these challenges, there is an urgent need for a predictive model that can effectively handle multiple feature types, dynamically identify vital features, and be well interpretable. This study proposes the FMI-CAECD model, a novel approach combining a deep feature extractor with an ensemble learning predictor. The manuscript’s contributions are as follows:This study proposes a multi-input 1D-CNN architecture for feature extraction in CVD prediction. This architecture facilitates the concurrent processing of various data modalities (e.g., numerical, ordinal, and categorical). Compared to traditional single-input CNN, this approach captures more complex interrelationships between different data types, potentially enhancing feature extraction efficacy and overall model performance.To guide the model’s focus on informative physiological features, this study incorporates a channel attention mechanism within the deep feature extractor. This mechanism dynamically allocates weights to feature channels, prioritizing those most relevant to the prediction task. The model effectively filters out noise and irrelevant information by focusing on critical physiological indicators, resulting in a more accurate and robust prediction.This study additionally incorporates a multi-model fusion strategy to potentially enhance the model’s generalizability to new or unseen data. Single models can be prone to bias and overfitting, especially when dealing with unbalanced or small datasets. Ensemble learning leverages the strengths of multiple models, improving prediction accuracy and robustness, particularly for rare disease detection.By incorporating SHAP analysis, this study pioneers a novel approach to interpreting deep learning models in clinical settings. SHAP assigns an attribution value to each feature, quantifying its influence on the model’s prediction for a specific instance. By enhancing model interpretability, we improve trust in its predictions and uncover critical CVD risk factors, providing invaluable clinical insights.

The remainder of this paper is structured as follows. Section 2 presents a comprehensive review of the existing literature on CVD prediction using ML techniques. Section 3 details the research methodology employed in this study. Section 4 outlines the dataset utilized, the experimental setup adopted, and the performance evaluation metrics. Finally, Section 5 concludes the paper by summarizing the essential findings and outlining potential future research directions.

## 2. Related Work

This section provides a comprehensive overview of prior research on CVD prediction using ML and DL techniques. A substantial body of prior work has investigated the use of ML for CVD prediction. However, the emergence of DL has led to a growing number of studies demonstrating its potential to outperform traditional ML methods in various aspects of CVD prediction.

The following subsections review vital studies from both ML and DL perspectives, focusing on their strengths and limitations. This analysis aims to identify the gaps and challenges in existing research that call for further advancements in CVD prediction models.

### 2.1. Machine Learning

A recent study [3] evaluated the performance of seven machine learning models for coronary heart disease (CHD) prediction. The models included LR, SVM, deep neural network (DNN), decision tree (DT), naive Bayes (NB), RF, and K-nearest neighbor (K-NN). Their study found that the DNN achieved the highest accuracy (98.15%), along with sensitivity (98.67%) and precision (98.01%). Despite their impressive performance, DNN models often lack interpretability, hindering clinicians’ understanding and trust in their predictions. Additionally, their sensitivity to data imbalance can limit performance when predicting rare events.

Another study [4] proposed a heart disease prediction model that combines feature selection and principal component analysis. By combining chi-square selection and RF, they extracted key features from the UCI heart dataset and achieved high-accuracy heart disease classification. This method offers advantages in reducing data dimensionality and potentially improving model performance, particularly when handling large datasets. The model achieved high accuracy through effective management of data imbalance and outliers. Despite its effectiveness, the model may have limitations in capturing complex interrelationships between different feature types, potentially hindering its predictive capabilities in heterogeneous datasets.

A heart disease prediction model (HDPM) was developed in a recent study [5], incorporating several data preprocessing techniques to potentially enhance model performance. This approach leverages DBSCAN for outlier detection and removal within the data. Subsequently, it employs SMOTE-ENN to address potential class imbalances within the training data distribution. Finally, the model utilizes an XGBoost classifier to predict heart disease. The authors report that their model achieved high accuracy on a public heart disease dataset. While effective, this model heavily relies on multiple preprocessing steps, which can introduce complexity and increase computational overhead.

A hybrid machine learning technology (HRFLM) that combines RF with a linear model (LM) for heart disease prediction was introduced in [6]. This approach leverages the strengths of both models, potentially achieving high prediction accuracy while enhancing model interpretability compared to traditional single models. However, the model’s reliance on conventional ensemble learning techniques may limit its capacity to dynamically prioritize critical features based on their physiological relevance.

Another recent study [7] proposed a machine learning-based hybrid decision support system for early heart disease detection. The system addressed missing data using a multivariate interpolation chain equation and employed a hybrid method combining a genetic algorithm with recursive feature elimination for feature selection. The study compared several classifiers, including SVM, NB, LR, RF, and AdaBoost. Their findings indicated that the random forest classifier achieved the highest accuracy, suggesting its potential as an effective tool for early heart disease detection. While hybrid approaches offer potential benefits, they may not effectively leverage the interactive relationships between different feature types.

### 2.2. Deep Learning

A cardiovascular disease prediction model using an improved deep belief network (DBN) architecture was introduced in [8]. This approach combines unsupervised pre-training with supervised fine-tuning, potentially enhancing model stability and prediction accuracy. The study reports a classification accuracy of 91.26% on the test dataset, suggesting the effectiveness of the DBN model for cardiovascular disease prediction. While the DBN demonstrates effectiveness in specific applications, its reliance on unsupervised pre-training limits its ability to prioritize relevant features during supervised training dynamically.

A two-layer CNN architecture for classifying CHD in imbalanced clinical datasets was proposed in [9]. Their model achieved an accuracy of 77%, demonstrating effectiveness in CHD classification, particularly for non-CHD cases. The authors compared their CNN approach to traditional machine learning methods like SVM and random forest, reporting improved balanced accuracy and overall performance. However, these models typically process only a single type of data, limiting their effectiveness.

A dual fully connected neural network (DFCNN) approach for high-precision arrhythmia classification was explored in [10]. Their method involved extracting 105 features from the data and utilizing a two-layer classifier to achieve accurate heartbeat classification. The study reported high sensitivity and accuracy in arrhythmia detection, particularly for identifying rare supraventricular premature beats (S-type). However, the model’s reliance on manual feature engineering and fully connected layers may hinder its ability to identify the most salient features dynamically.

An automated heartbeat classification system using a combined CNN and LSTM approach for arrhythmia detection in ECGs was developed in [11]. Their system combined a CNN and a long short-term memory network (LSTM) to achieve real-time ECG diagnosis. The study reported an accuracy of 94.20%, suggesting the effectiveness of this deep learning approach for ECG-based arrhythmia classification. Although practical, this approach focuses primarily on a single feature type, limiting the ability to prioritize critical features based on clinical significance dynamically.

A deep learning model combining CNN and LSTM with a dual attention mechanism for ECG-based arrhythmia classification, was presented in [12]. This approach leverages the strengths of both CNN and LSTMs: CNN for capturing spatial features and LSTM for capturing temporal information in ECG data. The dual attention mechanism potentially enhances the interpretability of the model. The study achieved an accuracy of 98.51% on the AFDB dataset, suggesting the effectiveness of this approach for arrhythmia detection. While effective in capturing temporal and spatial dependencies, this method may not fully leverage the intricate relationships between diverse feature types.

Another study [13] explored a deep learning model for heart disease prediction combining CNN and gated recurrent units (GRUs). Their approach utilized LDA and PCA for feature extraction from the data. The CNN-GRU model achieved an accuracy of 94.5%, exceeding the performance of LSTM (92.0%) and CNN-LSTM (93.7%) models on the same dataset. As effective as this method is, it relies heavily on manual feature selection methods and may overlook the implicit relationships in the data.

A hybrid deep learning model for heart disease prediction incorporating feature selection and a deep neural network was proposed in [14]. Their approach utilizes a Genetic Sinusoidal Algorithm (GSA) for feature selection, aiming to optimize the feature set and potentially improve model performance. The selected features are then fed into a DPA-RNN+LSTM model for disease prediction. This method potentially reduces computational cost by eliminating redundant features and achieves high prediction accuracy. This model is effective, but feature selection using a genetic algorithm can lead to significant computational overhead.

A novel heart disease prediction framework utilizing a Clustered Bidirectional Long Short-Term Memory (C-BiLSTM) network was presented in [15]. The C-BiLSTM model, leveraging bidirectional recurrent neural networks, effectively captured temporal dependencies and long-term memory, outperforming traditional methods on the UCI heart disease dataset. C-BiLSTM excelled in handling imbalanced datasets, making it a valuable tool for early heart disease detection. Despite its effectiveness in capturing time-dependent dependencies, C-BiLSTM may not fully handle the intricate relationships between various feature types.

A novel hybrid deep learning architecture for early disease risk prediction was proposed in [16]. The model synergistically integrates a Genetic Algorithm for feature selection, Stacked Autoencoders for dimensionality reduction, and a Softmax classifier for risk prediction. Comparative evaluations against traditional machine learning algorithms, including KN, DT, SVM, and CNN, demonstrated the superior performance and generalization capabilities of the proposed hybrid model. In spite of its effectiveness, the model’s reliance on stacked autoencoders and genetic algorithms introduces additional complexity and computational costs.

A novel hybrid deep learning architecture, MDenseNet201-IDRSNet, for early diagnosis of cardiovascular disease was introduced in [17]. MDenseNet201, with its dense connections, efficiently extracts both low-level and high-level features while reducing trainable parameters. Relief and LASSO algorithms select the most discriminative features, followed by IDRSNet for classification. MDenseNet201-IDRSNet achieved a remarkable prediction accuracy on UCI datasets, outperforming existing methods. Although this approach is practical, its multi-step feature selection process may limit flexibility and adaptability.

## 3. Method

This study investigates a novel CVD prediction model, FMI-CAECD, which utilizes a multi-input 1D-CNN architecture coupled with EL techniques (see Figure 1). The FMI-CAECD model architecture comprises three key stages: Feature Extraction, Feature Enrichment, and Ensemble Prediction module. In the Feature Extraction Module, the model employs tailored CNN architectures to effectively extract features from various data types, including numerical, ordinal, and categorical data. The Feature Enrichment Module refines the feature representation by employing global average pooling and a channel attention mechanism. The Ensemble Prediction Module comprehensively utilizes the advantages of three algorithms, XGBoost, LightGBM, and RF, to improve the accuracy and stability of prediction through multi-model fusion.

In addition, to gain further insights into the relationship between features and CVD risk, this study additionally employed SHAP analysis. SHAP analysis evaluates the impact of feature permutations on model performance, providing a means to interpret model predictions and potentially identify key CVD risk factors. This comprehensive approach contributes to potentially improved model performance and enhanced interpretability. Understanding these relationships can inform future research on CVD prevention and treatment strategies.

### 3.1. Feature Extraction Module

Considering the varied data types and feature structures, this study adopted a multi-input 1D-CNN architecture to process health data. Processing health data for cardiovascular disease prediction can be challenging due to the varied data types and feature structures. A 1D-CNN offers a promising approach for this task [18]. The 1D-CNN can automatically extract local features from one-dimensional data, potentially reducing the need for extensive preprocessing and manual feature engineering.

Furthermore, the multi-input 1D-CNN architecture allows processing different data types through tailored network branches. Each branch utilizes a 1D-CNN potentially optimized for the corresponding input type (e.g., numerical data, categorical data, ordinal data). This design can potentially improve the model’s flexibility in handling various data sources and enhance the overall expressiveness of the model. By optimizing the network architecture and parameters for each type of data, full utilization of each data dimension is ensured. The advantage of this multi-input 1D-CNN architecture lies in its high sensitivity to the intrinsic relationships within the data and its effective capability to learn complex patterns in health data. The structure of the 1D-CNN primarily consists of convolutional layers, activation functions, pooling layers, and fully connected layers.

The convolutional layer serves the purpose of feature extraction. It consists of a fixed number of filters (convolutional kernels) of a fixed size, each of which slides over the input data or the output from the previous layer, performing convolution operations to extract local features. The mathematical formulation of our model is given by
(1)$$x_{j}^{k}=\sum_{i\in M_{j}}w_{ij}^{k}x_{i}^{\left(k-1\right)}+b_{j}^{k}$$ here, $x_{j}^{k}$ denotes the input vector of the convolutional layer; $M_{j}$ denotes the size of the sensory field; and $w_{ij}^{k}$ and $b_{j}^{k}$ are two trainable parameters, which are the weight and bias of the convolutional kernel, respectively.

Activation functions introduce non-linearity, enabling the network to learn more complex feature representations. Considering the gradient vanishing problem, the rectified linear unit (ReLU) is chosen as the activation function. The mathematical formulation of our model is given by
(2)$$f\left(x\right)=\left\{\begin{array}{cc} x, & \mathrm{if}\;x>0\\ 0, & \mathrm{otherwise} \end{array}\right.$$
where *x* refers to the output of the convolutional layer.

The pooling layer is mainly used to reduce data dimensionality and enhance the model’s generalization capabilities. The proposed model employs two pooling strategies: Max Pooling and Global Average Pooling. The Max Pooling layer reduces feature dimensions by extracting the maximum value within the covered area. The Global Average Pooling layer simplifies each feature map into a single numerical value by calculating the average of the entire feature map.

The fully connected layer is located at the end of the network. It synthesizes the learned local features into global features and maps them to the output space. Each neuron in the fully connected layer is directly connected to all inputs, and can capture global patterns in the data.

#### 3.1.1. Numerical Data Processing Branch

In building deep learning models for CVD risk prediction, numerical data (e.g., metrics such as height, weight) contain a wealth of health information critical for accurately predicting CVD risk. To this end, we designed a branch of the network that specializes in this type of data, and the branch structure includes a one-dimensional convolutional layer, a ReLU activation function, and a max pooling layer. The one-dimensional convolutional layer is the core part of feature extraction. We employ 16 convolutional kernels of size 3, which effectively explore and extract local patterns and relationships within health data.

After the numerical data processing branch, the output feature vectors will become more compact and informative, effectively reflecting the interactions among health indicators and their combined effects on CVD risk. This highly abstract feature representation enhances the model’s depth of understanding of the data and improves the accuracy of predictions. Moreover, by processing features in this manner, the model can reduce the risk of misjudgments due to feature redundancy, enhancing the reliability and practicality of the model.

#### 3.1.2. Categorical Data Processing Branch

The Categorical Data Processing branch processes categorical data related to an individual’s health status including gender, disease history, etc. The architecture of this branch mainly contains 3 convolutional layers. The first convolutional layer uses 32 kernels of size 3 to capture primary relationships and patterns. In comparison, the second and third convolutional layers employ 64 and 128 kernels, to further deepen the network’s hierarchy. Each layer is equipped with batch normalization, ReLU activation functions, and a max pooling layer. The ReLU activation function and batch normalization layer are used to improve the speed and stability of model training. Max pooling layers reduce data dimensionality, thereby abstracting higher-level features.

The design of multiple convolutional and pooling layers aims to incrementally extract and abstract features, with each layer’s depth significantly enhancing the network’s ability to capture complex and deep patterns in the data. Overall, this multi-layer convolutional structure enables the categorical data processing branch to efficiently extract critical features from high-dimensional data, significantly improving the accuracy of predictions of CVD risk.

#### 3.1.3. Ordinal Data Processing Branch

In order to process and extract the features of ordinal data types like health status, smoking status, age group, etc., we have designed a specialized branch for ordinal data processing. This network architecture configures an embedding layer for each ordinal feature. Mapping each category to a low-dimensional space is critical for understanding essential features, such as health status, affecting CVD risk. A splicing layer is immediately used to combine the embedding outputs of all features into a comprehensive feature vector. This vector is then fed into a series of one-dimensional convolutional layers. Specifically, the first one-dimensional convolutional layer uses 32 convolutional kernels of size 3 to capture local features, followed by a max pooling layer of size 2. This process is repeated in the second layer of one-dimensional convolution, which uses 64 kernels to enhance the model’s feature extraction capability further.

The use of this structure allows the network to not only process input data of various lengths and scales, but also to extract complex and nonlinear patterns from these data that contribute to CVD risk assessment. Overall, the design of the Ordinal Data Processing Branch optimizes the processing of ordinal data, which significantly enhances the model’s predictive accuracy and generalization capabilities in tasks related to predicting CVD.

### 3.2. Feature Enrichment Module

Integrating information from multiple data sources is crucial in research on predicting CVD Risk. An innovative Feature Enrichment Module has been designed to amalgamate features from the Numerical Data Processing Branch, Categorical Data Processing Branch, and Ordinal Data Processing Branch efficiently. This module aims to enhance the overall performance and accuracy of the prediction model. A Global Average Pooling layer processes features extracted through the multi-input one-dimensional convolutional neural network, simplifying complex multi-dimensional features into one-dimensional vectors to reduce model complexity.

Although the multi-input 1D-CNN effectively extracts data features, not all features are equally important for predicting CVD risk. An innovative channel attention mechanism has been introduced to enhance the model’s sensitivity to key features. The channel attention mechanism dynamically adjusts feature weights, optimizing the model’s focus on key features, thereby improving the accuracy of predictions. This attention mechanism processes the concatenated comprehensive feature vector F, which has dimensions [Batch size, C], where C represents the total number of feature channels. Initially, a global average pooling layer generates feature descriptors for each channel, which are then processed through two fully connected layers. The first fully connected layer is responsible for dimension reduction, which helps decrease the model’s parameters and computational complexity. The second fully connected layer restores the original dimensions and outputs the importance weights for each channel through a sigmoid activation function. These weights are multiplied by the feature vector processed through the global average pooling layer, refining the feature intensity of each channel via element-wise multiplication. This produces the final optimized feature vector, which is used for subsequent prediction tasks.

Through this attention mechanism, the model can learn and enhance the intensity of feature channels beneficial for prediction while suppressing those that are not important. The design of this module thoroughly considers the diversity and complexity of medical data, ensuring that while maintaining efficiency, the model better captures and utilizes the key factors affecting CVD risk, providing solid data support for clinical decision-making.

### 3.3. Ensemble Prediction Module

Through the Feature Enrichment Module, we have successfully integrated complex features from multiple data sources, such as numerical, categorical, and ordinal data, generating a comprehensive feature vector that provides a rich information base for further prediction. The Ensemble Prediction Module incorporates RF, XGBoost, and LightGBM, selected due to their demonstrated superiority in a comparative experiment (Section 4.3.2). These models excel in handling complex feature interactions and consistently outperformed other methods in our study. Combining these models, the Ensemble Prediction Module harnesses their complementary strengths and mitigates the risk of biases or overfitting associated with individual models. The following sections describe the selected models (RF, XGBoost, and LightGBM).

#### 3.3.1. RF

RF is an ensemble classifier composed of many decision trees, analogous to a forest being a collection of many trees [19]. Each tree is constructed by randomly drawing samples from the original dataset through bootstrap sampling, ensuring diversity in the training set. The training process of each tree is independent, and other trees do not influence its results. RF is predicted by voting on the prediction results of all trees, using the Majority Voting (MV) mechanism to determine the final category. The mathematical formulation of our model is given by
(3)$${\mathrm{RaFo}}_{c}\left(p\right)=\mathrm{MV}{\left\{C_{q}\left(p\right)\right\}}_{1}^{Y}$$
where $C_{q}\left(p\right)$ is the categorization prediction of sample *p* by the *q*-th decision tree; MV is the majority vote of the constructed decision tree.

#### 3.3.2. XGBoost

XGBoost is an efficient machine learning algorithm based on Gradient-Boosted Decision Trees [20]. Its main characteristics include excellent system performance and rapid execution speed. The working principle of XGBoost involves progressively building the model, with each step attempting to correct the errors from the previous step. Specifically, in each iteration, XGBoost adds a new decision tree, aiming to correct the residuals of all previous trees. The mathematical formulation of our model is given by
(4)$${\hat{y}}_{i}^{\left(t\right)}={\hat{y}}_{i}^{\left(t-1\right)}+\eta \cdot f_{t}\left(x_{i}\right)$$
where ${\hat{y}}_{i}^{\left(t\right)}$ is the predicted value for the *i*-th sample after the *t*-th iteration; $f_{t}$ is the tree added in the *t*-th iteration; η is the learning rate; $x_{i}$ is the feature vector.

#### 3.3.3. LightGBM

LightGBM is an efficient learning algorithm based on the gradient boosting framework, demonstrating high efficiency and effectiveness when handling large-scale data [21]. It is particularly suitable for classification and regression tasks involving high-dimensional data. A distinctive feature of LightGBM is its use of a histogram-based decision tree algorithm, which utilizes histograms for node splitting to reduce memory usage and accelerate computation. Additionally, LightGBM incorporates a leaf-wise growth strategy, which can more effectively minimize the risk of overfitting than traditional depth-wise growth strategies.

#### 3.3.4. Model Ensemble

We employ a soft voting mechanism for the model ensemble to enhance the accuracy and stability of predictions. Soft voting aggregates the probability distributions from the base models, allowing for a more informed final decision.

Under this mechanism, each classifier $C_{i}$ outputs a prediction probability for each possible class label. Specifically, classifier $C_{1}$ outputs probabilities $P_{1}\left(L_{0}\right)$ and $P_{1}\left(L_{1}\right)$, corresponding to labels $L_{0}$ and $L_{1}$, respectively. Similarly, classifiers $C_{2}$ and $C_{3}$ generate corresponding probabilities $P_{2}\left(L_{0}\right)$, $P_{2}\left(L_{1}\right)$ and $P_{3}\left(L_{0}\right)$, $P_{3}\left(L_{1}\right)$. We calculate the composite prediction probability for each label through our EL strategy. The mathematical formulation of our model is given by
(5)$$P\left(L_{i}\right)=\arg\max_{c}\sum_{n=1}^{3}w_{n}\cdot P_{n}\left(L_{i}\right)$$
where $P_{n}\left(L_{i}\right)$ denotes the prediction result of the nth model; *i* takes the value of 0 or 1 to indicate whether it is diseased or not; and $w_{n}$ is the weight assigned to the nth model (these weights can be adjusted according to the performance of the model, here $w_{n}$ = 1/3).

The final category prediction selects the label with the highest integrated probability (see Figure 2). For example, if $P\left(L_{0}\right)>P\left(L_{1}\right)$, the final predicted category is $L_{0}$; conversely, it is $L_{1}$. This approach allows the integrated learner to utilize the predictive power of multiple models, thereby improving the accuracy and reliability of the overall prediction.

### 3.4. Risk Factor Screening

To identify potential risk factors for CVD patients, we utilized SHAP values to assess the importance of each feature. SHAP values represent an advanced method for interpreting models based on the Shapley values from game theory. They quantify the contribution of various potential risk factors to the predictive outcomes of diseases.

Given a set of features, we first utilize a trained deep learning model to make predictions regarding cardiovascular conditions, and select a representative subset of the data. We then use the DeepExplainer tool from the SHAP library to calculate the SHAP values for each sample in this subset. These values can be interpreted as each feature’s positive or negative impact on the model’s predictive output. The mathematical formulation of our model is given by
(6)$$\phi_{j}=\sum_{S\subseteq N\setminus \{j\}}\frac{|S|!(|N|-|S|-1)!}{|N|!}\left[f_{S}\left(x\right)-f_{S\setminus \{j\}}\left(x\right)\right]$$
where $\phi_{j}$ represents the SHAP value for feature *j*, *N* is the set of all features, and *S* is any subset of *N* excluding feature *j*. |S| denotes the number of features in subset *S*, and |N| is the total number of features. $f_{S}\left(x\right)$ is the model *f*’s prediction output for input *x* under the condition that subset *S* is included, while $f_{S\setminus \{j\}}\left(x\right)$ is the prediction output when subset *S* does not include feature *j*.

Through SHAP values, we can identify features that significantly influence the risk of cardiovascular disease, and explore how these features impact disease prediction. This facilitates a more comprehensive grasp of the potential risk factors.

### 3.5. Performance Metrics

Model evaluation is crucial as it provides insights into the model’s performance, strengths, and weaknesses. Therefore, various evaluation metrics widely used in the literature are employed in this study to assess the proposed model. These metrics include Accuracy, Sensitivity, Specificity, F1-Score, and AUC. The mathematical formulation of our model is given by
(7)$$\mathrm{Accuracy}=\frac{\mathrm{TN}+\mathrm{TP}}{\mathrm{TN}+\mathrm{FN}+\mathrm{TP}+\mathrm{FP}}$$
(8)$$\mathrm{Sensitivity}=\frac{\mathrm{TP}}{\mathrm{TP}+\mathrm{FN}}$$
(9)$$\mathrm{Specificity}=\frac{\mathrm{TN}}{\mathrm{TN}+\mathrm{FP}}$$
(10)$$\mathrm{F1-Score}=\frac{2\cdot \mathrm{TP}}{2\cdot \mathrm{TP}+\mathrm{FP}+\mathrm{FN}}$$
(11)$$\mathrm{AUC}=\frac{1}{2}\left[\frac{\mathrm{TP}}{\mathrm{TP}+\mathrm{FN}}+\frac{\mathrm{TN}}{\mathrm{TN}+\mathrm{FP}}\right]$$

TP refers to the count of samples for which the model accurately predicts the presence of cardiovascular disease. FP denotes the number of individuals incorrectly predicted by the model to have cardiovascular disease despite not having it. TN is the sample correctly identified by the model as not having cardiovascular disease. FN represents the instances where the model erroneously predicts individuals with cardiovascular disease do not have the condition.

## 4. Experiment

### 4.1. Datasets

This study utilized data from the Behavioral Risk Factor Surveillance System (BRFSS) 2022 [22]. BRFSS is an extensive survey conducted by the Centers for Disease Control and Prevention (CDC) in the United States. It gathers data on health-related behaviors and risk factors among the adult population. The survey conducted telephonic interviews with adults across all states of the United States, gathering data on a variety of health-related behaviors, including smoking, alcohol consumption, healthcare utilization, mental health status, obesity prevalence, and exercise habits. The dataset has a total of 445,132 records and 326 features.

The BRFSS dataset relies on self-reported information collected through telephonic interviews, which provides comprehensive coverage of the adult population across the United States. While self-reported information may not always capture undiagnosed conditions, such as CVD, the dataset remains a robust and reliable source for population-level health assessments. The potential impact of such undiagnosed cases is considered minimal about the dataset’s overall predictive value. While this limitation is common in survey-based data, the BRFSS still provides valuable insights into CVD risk factors across the population.

#### 4.1.1. Feature Selection

From the initial dataset consisting of 326 features, we performed an in-depth feature selection based on a combination of literature review and empirical testing. Following a comprehensive review of the relevant literature [23,24,25,26], we identified a set of variables consistently reported as significant predictors of CVD and related health outcomes. To refine our feature set, we employed Lasso regression to identify the most relevant variables by shrinking the coefficients of less essential features. Lasso regression was selected due to its ability to handle high-dimensional data and prevent overfitting through regularization, making it ideal for our dataset with numerous initial features. This hybrid approach enabled us to refine our feature set to 40 critical variables deemed to have substantial predictive power in our study.

These selected features are categorized into numerical, categorical, and ordinal. Numerical features represent continuous quantitative measurements that can take a wide range of values (see Table 1). Categorical features represent distinct groups or classes without an inherent order (see Table 2). On the other hand, ordinal features capture ordered categories where the sequence or ranking of the values is meaningful (see Table 3).

These features were preprocessed using appropriate techniques, including reshaping and embedding, to ensure compatibility with our multi-input model architecture. This comprehensive approach to feature selection allowed us to build a predictive model that integrates a diverse set of variables, reflecting various aspects of cardiovascular risk.

#### 4.1.2. Imbalance Analysis

High-quality data are essential for developing accurate and reliable prediction models [27]. The telephone survey methodology employed in BRFSS 2022 data collection may have resulted in a higher prevalence of missing values compared to other data collection methods. Additionally, a significant portion of respondents either declined to participate in specific questions, or expressed uncertainty regarding their responses. After implementing rigorous data cleaning procedures to address missing and rejected inputs, the resulting dataset comprises 246,016 high-quality records. A meticulous analysis of the dataset reveals that the distribution of healthy individuals compared to those with cardiovascular disease within the target variable stands at 94.5% and 5.5%, respectively. This discrepancy underscores the pronounced imbalance in the target variable of our dataset. To mitigate the issue of class imbalance within our dataset, we employed oversampling techniques to equalize the distribution between classes. This strategic adjustment was aimed at enhancing our model’s predictive accuracy and overall performance (see Figure 3).

In addition to addressing the class imbalance, we conducted an in-depth analysis of the distribution of the 40 selected features to identify potential imbalances that may introduce bias into our model predictions. For numerical characteristics, we calculated descriptive statistics, including the mean, median, standard deviation, skewness, and kurtosis, to assess whether the distribution of these characteristics showed significant deviations. For example, AgeCategory showed a higher proportion of individuals aged above 60, which aligns with the typical age distribution among individuals at higher risk for CVD. Similarly, TotalCholesterol values exhibited a positively skewed distribution, with a considerable portion of the dataset having elevated cholesterol levels, reflecting the known association between high cholesterol and CVD risk.

Frequent distributions were examined for categorical and ordinal features to identify any categories that were disproportionately represented. Notably, features such as SmokerStatus revealed a higher prevalence among CVD patients, which is consistent with established risk factors. These natural imbalances are expected and align with real-world health disparities. Therefore, instead of artificially balancing these features, we preserved their inherent distributions to maintain the validity of the relationships between these features and CVD risk.

We conducted sensitivity analyses by comparing model performance under different data balancing scenarios to assess further the potential impact of feature imbalances on model predictions. Our findings indicated that retaining the natural distribution of these features led to more accurate and reliable predictions, reinforcing the importance of maintaining these natural imbalances.

#### 4.1.3. Correlation Analysis

Additionally, we analyzed the interactions between the dataset features to identify potential correlations. This examination was essential to understanding the complex relationships that could influence the model’s predictive capabilities. If multicollinearity exists among the variables, the model may produce unreasonable results. Therefore, it is necessary to identify and remove variables that exhibit strong correlations (see Figure 4).

This figure presents a heatmap that illustrates the positive/negative correlations among the features. Each cell C(i,j) in the grid represents the correlation of the features in the *i*-th row and *j*-th column. The heatmap displays the strength of correlations between features through varying colors in its cells. Colors closer to yellow indicate stronger correlations (either positive or negative), while colors closer to purple denote weaker correlations. From Figure 4, it can be observed that all correlation coefficients are below 0.6, indicating the absence of highly correlated features. However, one feature pair exhibits a correlation of 0.69, which suggests a moderate correlation. We carefully assessed this feature pair and determined that, despite the moderate correlation, both features provide unique and valuable information to the model. Consequently, all these features will be input into the proposed method.

The Spearman correlation coefficient is used here to measure the correlation between features. This method applies to continuous data and ordinal categorical data, effectively handling non-normal distributions and outliers within the dataset. The Spearman’s rank correlation coefficient ranges from −1 to 1, where 1 indicates a perfect positive correlation, −1 indicates a perfect negative correlation, and 0 signifies no correlation. The mathematical formulation of our model is given by
(12)$$r_{s}=1-\frac{6\sum d_{i}^{2}}{n(n^{2}-1)}$$ here, $d_{i}$ represents the difference in ranks of the two variables at the *i*-th data point, and *n* is the total number of data points.

### 4.2. Experimental Setup

To avoid randomness, all experiments were subjected to 10-fold cross-validation. The training of the models was conducted using the ADAM optimizer, which combines the advantages of momentum techniques and adaptive learning rate adjustment, which is particularly effective in large-scale datasets and high-dimensional parameter spaces. Due to its rapid optimization capabilities, in-variance to gradient re-scaling, and the potential to utilize sparse gradients, significantly enhance the performance of neural networks. The chosen loss function is binary cross-entropy. This function measures the discrepancy between the model’s predicted probability distribution and the actual distribution of the target. The mathematical formulation of our model is given by
(13)$$L=-\frac{1}{N}\sum_{i=1}^{N}\left[y^{\left(i\right)}\log\left(p^{\left(i\right)}\right)+\left(1-y^{\left(i\right)}\right)\log\left(1-p^{\left(i\right)}\right)\right]$$
where $y^{\left(i\right)}\in \left\{0,1\right\}$ is the label, and $p^{\left(i\right)}=\sigma \left(f\left(x^{\left(i\right)}\right)\right)$ is the model’s predicted probability that the *i*-th sample is an upbeat class. This loss function aims to minimize the information entropy between the predicted probabilities and the actual labels. Optimizing this loss enables the model to accurately estimate the probabilities of events occurring and not occurring during predictions.

### 4.3. Experimental Results

To implement our method, the Python programming language and various commonly used libraries such as pandas, NumPy, Matplotlib, Seaborn, TensorFlow, Keras, and Scikit-learn were employed. The execution was performed in Jupyter Notebook, part of the Anaconda 3 distribution.

#### 4.3.1. Risk Factor Screening Results

To investigate which features contribute most significantly to the risk of CVD, this study employed SHAP values to identify the features that have a substantial impact on cardiovascular risk (see Figure 5).

The results indicate that the importance of “AgeCategory” is particularly significant, aligning with our expectations, as age is a significant risk factor for CVD. The risk of CVD significantly increases with age. “GeneralHealth” also plays a crucial role, reflecting an individual’s overall health status; generally, better health is associated with a lower risk of CVD, and vice versa. Other important factors include “SystolicBloodPressure” and “AlcoholDrinks”, which are well-known contributors to CVD risk, with high blood pressure and excessive alcohol consumption being established risk factors. Overall, these results support the perspective that CVD risk assessment must consider a comprehensive array of factors, including physiological, biological, and lifestyle elements.

#### 4.3.2. High-Performance Filtering Results

In this section, high-performance filtering is used to select the meta-learners selected as the meta-learners for the ensemble prediction module (see Table 4). This process involves comparing various algorithms, including RF, LR, KNN, XGBoost, and LightGBM. Ultimately, RF, XGBoost, and LightGBM were selected as the meta-learners due to their complementary strengths and high performance across multiple evaluation metrics.

RF is well-known for its ability to handle high-dimensional data and effectively reduce overfitting through bagging and feature randomness. XGBoost, on the other hand, employs gradient boosting to enhance prediction accuracy by iteratively minimizing errors. LightGBM, a lightweight and highly efficient boosting algorithm, excels at handling large datasets and imbalanced data distributions. The combination of these models in the ensemble strategy leverages their unique strengths, leading to improved overall performance.

The ensemble learning (EL) approach, which integrates these selected meta-learners, further enhances predictive accuracy. The EL method outperformed individual models in nearly all metrics, achieving an accuracy of 92.70%, a sensitivity of 91.98%, and an AUC score of 91.65%. This superior performance demonstrates the effectiveness of the ensemble strategy in combining the strengths of multiple algorithms to achieve more accurate and reliable CVD predictions. By leveraging the complementary characteristics of RF, XGBoost, and LightGBM, the EL approach optimally balances bias and variance, improving generalization on unseen data. This complementary nature helps in effectively capturing diverse data patterns and mitigating individual model limitations.

#### 4.3.3. Comparative Experiment Results

To assess the efficacy of our proposed model, we conducted a comparative analysis against several state-of-the-art models. The outcomes of these comparisons substantiate the validity and performance of our approach (see Table 5).

Compared to prior models, the proposed deep learning-based CVD prediction model outperforms significantly. Achieving an accuracy of 97.45%, our model significantly outperforms state-of-the-art approaches, including the C-BiLSTM model (93.57%), DPA-RNN+LSTM model (94.49%), MDenseNet201-IDRSNet model (94.70%), and GA-SAE-Softmax model (95.03%).

The C-BiLSTM model effectively captures temporal dependencies through its bidirectional LSTM architecture, which is beneficial for sequential data analysis. However, its reliance on sequential feature learning may limit its performance in handling complex structured datasets with multiple feature types. Similarly, the DPA-RNN+LSTM model incorporates a dual-path attention mechanism to enhance temporal feature extraction. However, its architecture is primarily focused on sequential data, and may not fully leverage the interrelationships between different types of features.

The MDenseNet201-IDRSNet model leverages dense connections to improve feature learning and information flow, achieving slightly higher accuracy. Despite its effectiveness in feature extraction, its complex structure may pose challenges regarding model interpretability. The GA-SAE-Softmax model combines genetic algorithms with stacked autoencoders for optimized feature selection and dimensionality reduction, improving accuracy. However, its reliance on autoencoders may make it sensitive to noise and high dimensionality in data.

In contrast, our proposed FMI-CAECD model incorporates a multi-input 1D-CNN architecture that enables the parallel processing of numerical, ordinal, and categorical data. This design effectively captures complex interrelationships between different feature types, enhancing feature extraction efficiency. Additionally, integrating a channel attention mechanism allows the model to dynamically focus on the most informative features, thereby improving predictive accuracy. This significant improvement validates the superiority of the FMI-CAECD model in CVD prediction and highlights its notable potential in enhancing model interpretability and performance.

#### 4.3.4. Ablation Experiment Results

Ablation experiments observe the specific contribution of each part to the overall predictive performance by systematically removing key modules from the structure. All experiments are conducted on the same dataset for both training and testing (see Table 6).

The results indicate that models incorporating attention mechanisms and EL outperform those using solely multi-input 1D-CNN across all evaluation metrics, including accuracy, sensitivity, specificity, F1 score, and AUC values. When an attention mechanism (CNN+Att) was added, the accuracy increased from 94.76% to 95.61%. This shows that the attention mechanism allows the model to assign higher weights to more informative features, thereby enhancing the feature representation and more effectively capturing key patterns. These results demonstrate the effectiveness and superiority of our proposed method. Especially when dealing with complex datasets, it is capable of delivering more accurate and robust performance.

## 5. Conclusions

CVD remains a leading cause of mortality despite significant advancements in medical technology. This paper proposes a novel approach to CVD prediction. It combines a deep feature extractor with an integrated predictor, aiming to achieve superior accuracy and early detection compared to existing methods. The architecture of the deep feature extractor predominantly employs an enhanced multi-input 1D-CNN tailored for processing diverse data types. This system integrates a channel attention mechanism to augment feature enrichment, enhancing the specificity and relevance of the extracted features. An advanced selection process utilizing high-performance filtering in the prediction segment identifies three superior meta-learners. This methodical selection is designed to optimize the accuracy and robustness of the integrated predictive model.

The proposed model’s efficacy is evaluated through a series of experiments conducted on the BRFSS 2022 dataset. The dataset comprises 246,016 records and 40 features. These data encompasses a broad range of health-related domains, including lifestyle behaviors, medical history, socioeconomic factors, and physical assessment findings. Compared to traditional approaches, our proposed method achieves competitive performance on the BRFSS 2022 dataset, demonstrating statistically significant improvement on established metrics such as accuracy, sensitivity, and AUC. We employ the SHAP approach for feature importance measurement to improve the model’s interpretability and gain insights into feature importance. Such transparency can facilitate informed decision-making for both clinicians and patients.

While the proposed models exhibit promising performance, a comprehensive evaluation requires acknowledging certain limitations. The current study primarily concentrates on structured tabular data, a common source for clinical decision-making. To broaden the applicability of these models, future research should investigate the integration of multimodal data, such as echocardiography or imaging modalities. Such an approach could provide complementary information, potentially enhancing the generalizability and robustness of the models.

Another limitation lies in the ensemble learning using multiple learners. While ensemble methods can enhance prediction accuracy and stability, they often introduce increased computational complexity and resource demands. This can hinder their practical implementation in real-time or large-scale clinical settings. Future research should explore strategies for optimizing ensemble methods to balance computational efficiency and predictive performance.

It should also be noted that the BRFSS dataset is based on self-reported information collected through telephonic interviews. As a result, there may be instances where individuals with undiagnosed CVD are not accurately captured. However, given the large sample size and comprehensive nature of the BRFSS data, the potential impact of such misclassification is considered minimal.

## Figures and Tables

**Figure 1 sensors-24-07160-f001:**
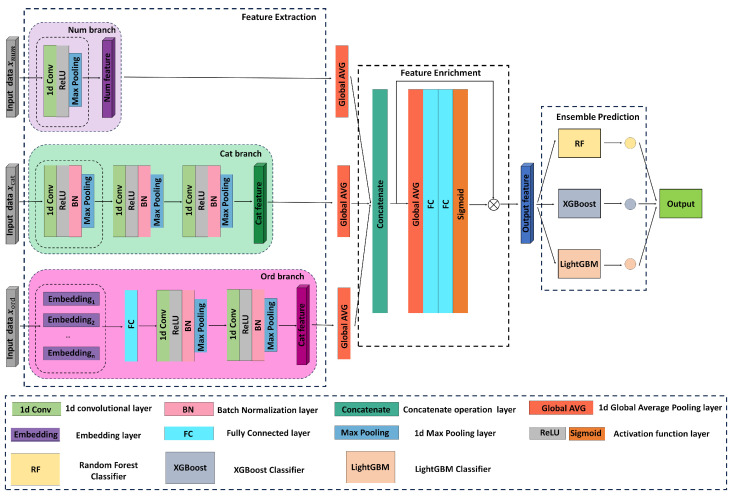
FMI-CAECD network architecture.

**Figure 2 sensors-24-07160-f002:**
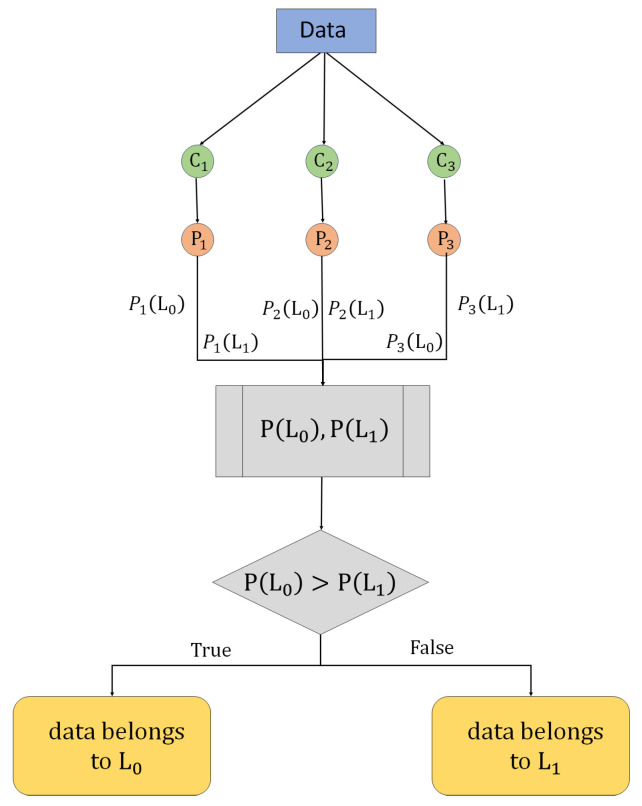
Soft voting classifier (ensemble technique).

**Figure 3 sensors-24-07160-f003:**
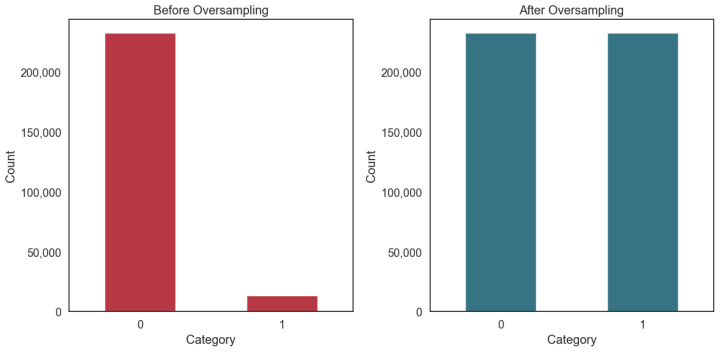
Comparison of oversampling results.

**Figure 4 sensors-24-07160-f004:**
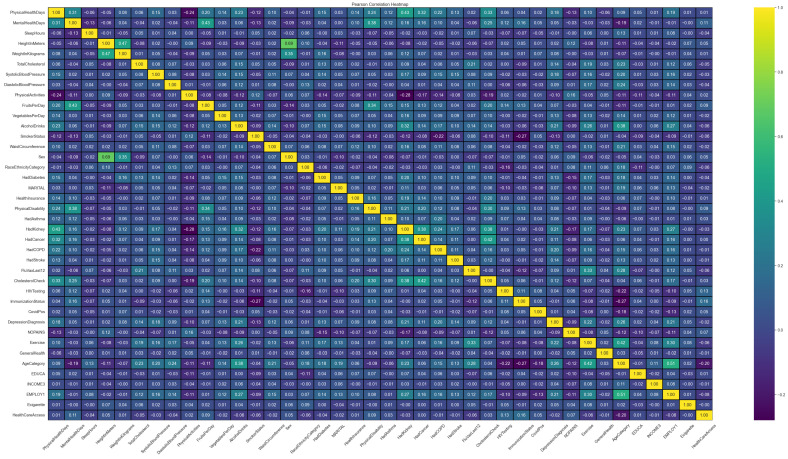
Characteristic correlation heat map.

**Figure 5 sensors-24-07160-f005:**
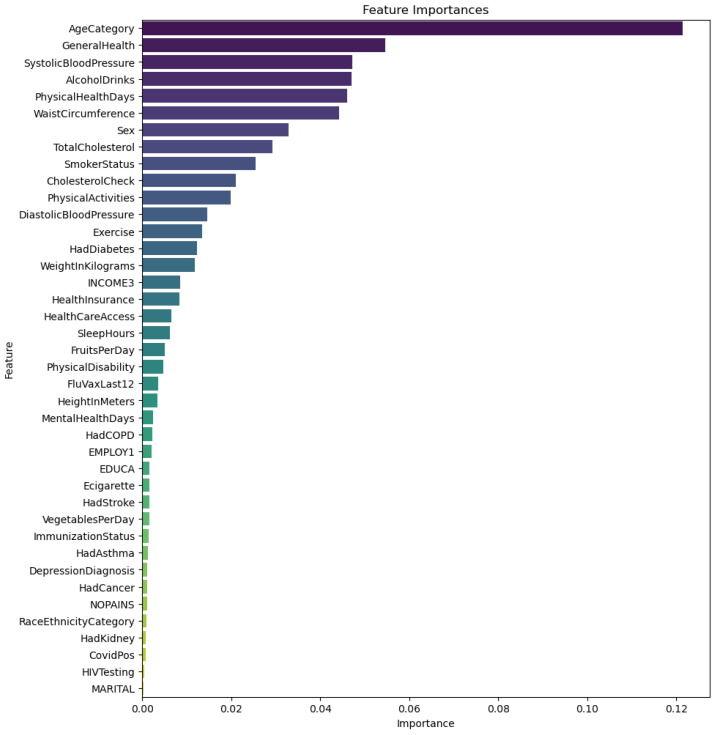
Importance ranking of features.

**Table 1 sensors-24-07160-t001:** Description of numerical features.

Feature Type	Feature Name	Description
Numerical	PhysicalHealthDays	Number of days with poor physical health
Numerical	MentalHealthDays	Number of days with poor mental health
Numerical	SleepHours	Average hours of sleep per night
Numerical	HeightInMeters	Height in meters
Numerical	WeightInKilograms	Weight in kilograms
Numerical	TotalCholesterol	Total cholesterol level
Numerical	SystolicBloodPressure	Systolic blood pressure (mmHg)
Numerical	DiastolicBloodPressure	Diastolic blood pressure (mmHg)
Numerical	PhysicalActivities	Minutes of physical activity per week
Numerical	FruitsPerDay	Servings of fruits per day
Numerical	VegetablesPerDay	Servings of vegetables per day
Numerical	AlcoholDrinks	Number of alcoholic drinks per week
Numerical	WaistCircumference	Waist circumference (cm)

**Table 2 sensors-24-07160-t002:** Description of categorical features.

Feature Type	Feature Name	Description
Categorical	Sex	Gender
Categorical	RaceEthnicityCategory	Race/ethnicity
Categorical	HadDiabetes	Diabetes status
Categorical	MARITAL	Marital status
Categorical	HealthInsurance	Health insurance status
Categorical	PhysicalDisability	Physical disability status
Categorical	HadAsthma	Asthma diagnosis
Categorical	HadKidney	Kidney disease diagnosis
Categorical	HadCancer	Cancer diagnosis
Categorical	HadCOPD	Chronic obstructive pulmonary disease diagnosis
Categorical	HadStroke	Stroke history
Categorical	FluVaxLast12	Received flu vaccine in the past year
Categorical	CholesterolCheck	Ever had cholesterol checked
Categorical	HIVTesting	At risk for HIV
Categorical	ImmunizationStatus	Immunization status
Categorical	COVIDPos	COVID-19 vaccination status
Categorical	DepressionDiagnosis	Diagnosis of depression
Categorical	NOPAINS	Reasons for physical inactivity
Categorical	Exercise	Physical activity level

**Table 3 sensors-24-07160-t003:** Description of ordinal features.

Feature Type	Feature Name	Description
Ordinal	GeneralHealth	Self-reported general health status
Ordinal	AgeCategory	Age category
Ordinal	EDUCA	Education level
Ordinal	INCOME3	Income level category
Ordinal	EMPLOY1	Employment status
Ordinal	SmokerStatus	Smoking status
Ordinal	Ecigarette	Electronic cigarette use status
Ordinal	HealthCareAccess	Frequency of accessing healthcare

**Table 4 sensors-24-07160-t004:** Results of high-performance filtering.

Method	Accuracy (%)	Sensitivity (%)	Specificity (%)	F1-Score (%)	AUC (%)
LR	85.56	84.19	85.93	84.11	85.37
KNN	85.70	84.98	87.40	85.50	85.85
XGBoost	85.92	86.27	88.56	85.87	86.33
LightGBM	86.98	86.13	87.84	86.96	88.35
RF	89.26	89.41	90.10	89.24	90.58
EL	92.70	91.98	90.07	92.50	91.65

**Table 5 sensors-24-07160-t005:** Results of comparative experiments.

Reference	Method	Year	Accuracy (%)
Dileep et al. [15]	C-BiLSTM	2023	93.57
Vai et al. [14]	DPA-RNN+LSTM	2023	94.49
Mandava et al. [17]	MDenseNet201-IDRSNet	2024	94.70
Bülbül et al. [16]	GA-SAE-Softmax	2024	95.03
Ours	FMI-CAECD		97.45

**Table 6 sensors-24-07160-t006:** Results of ablation experiments.

Method	Accuracy (%)	Sensitivity (%)	Specificity (%)	F1-Score (%)	AUC (%)
CNN	94.76	93.54	92.83	93.67	94.38
CNN+Att	95.61	94.23	91.40	92.51	95.85
Ours	97.45	96.84	95.07	92.44	96.68

## Data Availability

Data are publicly available on the CDC website at https://www.cdc.gov/brfss/index.html (accessed on 7 August 2024).

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
