# Peer review of "FMI-CAECD: Fusing Multi-Input Convolutional Features with Enhanced Channel Attention for Cardiovascular Diseases Prediction"

_sensors, 2024, doi:10.3390/s24227160_

Round 1
Reviewer 1 Report
Comments and Suggestions for Authors
In this manuscript, the authors developed a novel Framework for Multi-Input, One-dimensional Convolutional Neural Network(1D-6CNN) with Attention Mechanism for CVD (FMI-CAECD) for CVD risk assessment. Although this study presented some results, there are some important concerns to be addressed.
1. Rationale: The author should provide more background of the current CVD prediction models and the limitations of these models, and the needs of developing the novel model.
2. Do not use the proper names of authors in science papers when citing their work. The data, and not the personalities, is important.
3. The advantages of current prediction model should be discussed with other models.
4. Please provide the limitations of the current study and the potential solution.
Comments on the Quality of English Language
The language in the manuscript needs to be polished to improve readability of the manuscript.
Author Response
Comments 1: Rationale: The author should provide more background of the current CVD prediction models and the limitations of these models, and the needs of developing the novel model.
Response 1: Thank you for your constructive comments and insights. In response to your feedback, we have made changes to the introduction of the manuscript to better set the context of current CVD prediction models and the urgent need for progress in this area.
We have added detailed background information about existing CVD prediction models and analysed their limitations, in particular their inadequacy when dealing with structured data. Traditional machine learning models have obvious challenges in capturing complex feature relationships and coping with high-dimensional features and data imbalance, and existing deep learning methods also have limitations in fully utilising the interactive information of different feature types and the interpretability of the model.
To address these issues, we propose the FMI-CAECD model. This novel approach combines a deep feature extractor with an ensemble learning predictor, which can dynamically process multiple feature types, effectively identify key features and improve model interpretability. We believe that this model can meet the current needs of cardiovascular disease prediction by providing a more robust and clinically applicable solution.
For details of the specific changes, please refer to the detailed description in section 1.
Comments 2: Do not use the proper names of authors in science papers when citing their work. The data, and not the personalities, is important.
Response 2: Thank you for your suggestions regarding our citation specifications for papers. We recognise that there is indeed a lack of consideration in this area. In response to your suggestion of not using author names in citations, we have revised the manuscript accordingly. When citing previous studies, we have removed the author's name and replaced it with the name of the study's description, model, or method. This change is intended to ensure that the article is objective and scientific, focusing on the findings themselves. We believe this change meets the standards for scientific papers and makes the manuscript more rigorous and professional.
For detailed information, please refer to the detailed instructions in Section 2. Thank you again for your careful review of our work and for such constructive comments.
Comments 3: The advantages of current prediction model should be discussed with other models.
Response 3: Thank you for your helpful comments on our paper. In response to your suggestion, we have made the following changes to the manuscript:
(1) High-performance screening results: In this section, we compare the performance of several classical algorithms (including RF, LR, KNN, XGBoost, LightGBM, etc.) on different metrics. In the end, we chose RF, XGBoost, and LightGBM as the meta-learners for the integrated prediction module because they showed complementary advantages in multiple evaluation metrics. We explain the specific advantages of each algorithm in the text, such as RF's ability to handle high-dimensional data and reduce overfitting, XGBoost's mechanism for enhancing prediction accuracy through gradient boosting, and LightGBM's superior performance in handling large-scale data and imbalanced data. We also discuss how ensemble learning methods can improve overall performance and model generalisation by combining the complementary strengths of these models(see Section 4.3.2 for details).
(2) Comparing experimental results: In the ‘Comparative experimental results’ section of the revised manuscript, we provide a detailed comparison between the FMI-CAECD model and several other existing advanced models, including C-BiLSTM, DPA-RNN+LSTM, MDenseNet201-IDRSNet, GA-SAE-Softmax, etc. We explain the specific advantages of these models and their limitations. This in-depth analysis demonstrates the relative advantages of our model (see Section 4.3.3 for details).
(3)Ablation experiment results: We systematically verified the contribution of each module to the overall prediction performance of the model through ablation experiments, and added detailed analysis in the experimental section to explain the role of each key module in improving the accuracy and robustness of the model (see Section 4.3.4 for details).
These changes aim to respond to your comments and strengthen the discussion of the strengths and weaknesses of the existing model, so that the FMI-CAECD model we propose can be presented more comprehensively in the article. Thank you for your comments to help us improve the paper.
Comments 4: Please provide the limitations of the current study and the potential solution.
Response 4: Thank you for your insightful comments and feedback on our research. In response to your suggestions, we have revised the ‘Conclusions’ section to include a discussion of the study's limitations and future directions for improvement.
Specifically, we acknowledge two primary limitations in our study: data type constraints and model complexity. First, our study primarily focuses on structured tabular data, which is a valuable source for clinical decision-making. However, this focus may limit the model’s applicability to other medical data types, such as echocardiograms or imaging modalities. To enhance the model’s generalizability and robustness, we suggest that future research could explore integrating multimodal data sources.
Second, while ensemble learning methods significantly improve prediction accuracy and stability, they also increase computational complexity and resource requirements. This added complexity could pose challenges for real-time or large-scale clinical applications. Therefore, optimizing ensemble methods to achieve a better balance between computational efficiency and prediction performance is an important direction for future research.
We hope these revisions effectively address your suggestions, and we appreciate your valuable comments, which have helped to improve the quality of our paper.
Response to Comments on the Quality of English Language:
Thank you for your feedback on the quality of the English language. In response to your comments on the language and readability of the manuscript, we have carefully reviewed the entire text and made changes to improve clarity and consistency. Specifically, we have revised sentence structure to ensure clarity and simplicity of expression, standardised technical terminology for consistency, and conducted a thorough check of grammar and spelling.
In addition, we have reviewed the manuscript with a focus on readability and to ensure that key ideas are effectively communicated. If you have any further questions or suggestions in this regard, we would be grateful if you could let us know.
Reviewer 2 Report
Comments and Suggestions for Authors
In this work, the authors present interesting AI-approach for predicting cardiovascular diseases based on channel attention and fusion of multiple convolution inputs. Although the importance of creating a reliable predicting model for CVD, which is the theme of the manuscript, is out of questions, the quality of materials presenting (its consistency) should be improved. I have several concerns and comments for the authors, that are summarized below. I hope that they will help to improve the manuscript.
In this work, the authors present interesting AI-approach for predicting cardiovascular diseases based on channel attention and fusion of multiple convolution inputs. Although the importance of creating a reliable predicting model for CVD, which is the theme of the manuscript, is out of questions, the quality of materials presenting (its consistency) should be improved. I have several concerns and comments for the authors, that are summarized below. I hope that they will help to improve the manuscript.
Comments:
1. My main concern is the lack of clearly stated novelty of the proposed approach. In introduction section it should be clearly observed the problems of the state-of-the-art solutions and issues that the solution proposed by the authors is solving. Moreover, the novelty features of the present solution should be highlighted.
2. In section 2 ‘Related Work’ there is an overview of previously published works, but there is no analysis of their advantages and disadvantages, so that it is clear which of the problems have not been solved by existing solutions, what is the novelty of the proposed solutions compared to already available.
3. In subsection 3.3 there is no any explanation why these 4 specific models are described (RF, XGBoost, LightGBM, ME). Some logical connection in 3.3 subsection prior 3.3.1 is needed. Moreover, RF, XGBoost, LightGBM are both well known classical ML models, and providing such level of detail (as in 3.3.1..3.3.3) seems excessive.
4. P.10, line 388. ‘we proceeded with the selection of 40 variables for analysis’. How these 40 out of 326 features were chosen? The list in lines 389-391 contains less than 40 features; it would be advisable to add descriptions of these 40 features, preferably broken down into 3 categories used (numerical, categorical and ordinal).
5. Another question about the dataset used: how the presence or absence of the CVD was set for each data sample (response value used as ground truth). As the dataset contains telephonic interview data, there should be many of persons with undiagnosed CVD. If so, this limitation of the dataset and conclusions made using this dataset should be stated.
6. According to 402 there is class imbalance (healthy/with CVD), but there is no information about imbalance in 40 features used, it is worth to be analyzed also to provided information on some possible bias of the data that may have an impact on model prediction.
7. In conclusion section it is worth to mention the limitation of the proposed model and conclusion made by it.
Author Response
Comments 1: My main concern is the lack of clearly stated novelty of the proposed approach. In introduction section it should be clearly observed the problems of the state-of-the-art solutions and issues that the solution proposed by the authors is solving. Moreover, the novelty features of the present solution should be highlighted.
Response 1: Thank you for your careful review and constructive comments on our paper. We have carefully revised the manuscript to address your concerns about the clarity of the challenges in existing models and the novelty of our proposed approach.
We have added to the introduction the main challenges of current models when dealing with structured data, including insufficient capture of complex feature relationships, sensitivity to high-dimensional features and data imbalance, dependence on a single data input, and limited interpretability. These additions aim to highlight the limitations of existing models and emphasise the need to propose new models.
Based on the challenges faced by existing models, we propose an urgent need to construct a prediction model that can effectively handle multiple feature types, dynamically identify key features, and have good interpretability. On this basis, we propose the FMI-AECD model. In addition, we have specifically added a brief description of each innovative feature to better highlight the uniqueness of this study (see Section 1 for details).
We sincerely appreciate your feedback, which will help us improve the clarity and expressiveness of the manuscript.
Comments 2: In section 2 ‘Related Work’ there is an overview of previously published works, but there is no analysis of their advantages and disadvantages, so that it is clear which of the problems have not been solved by existing solutions, what is the novelty of the proposed solutions compared to already available.
Response 2: Thank you for your useful suggestions regarding the ‘Related work’ section. We have improved this section based on your suggestions. In the updated ‘Related work’ section, we have analysed in detail the advantages and limitations of each method, especially their performance in predicting CVD. In this way, we hope to clearly demonstrate the unresolved issues in existing research, thereby providing a reasonable background for our proposed solutions.
Also, since we have already described the contribution of our study in the introduction, we have focused on the analysis of existing methods in the ‘Related work’ section and have not repeated the introduction of our methods in that section. We believe that this change can improve the logical clarity and coherence of the article.
Thank you for your excellent feedback on our work, which has enabled us to further improve the paper.
Comments 3: In subsection 3.3 there is no any explanation why these 4 specific models are described (RF, XGBoost, LightGBM, ME). Some logical connection in 3.3 subsection prior 3.3.1 is needed. Moreover, RF, XGBoost, LightGBM are both well known classical ML models, and providing such level of detail (as in 3.3.1..3.3.3) seems excessive.
Response 3: Thank you for your important comments on the selection and description of the models in Section 3.3. We understand your suggestions regarding the content of Section 3.3 and have made changes based on the feedback.
At the beginning of Section 3.3, we have added a short paragraph explaining the reasons for introducing RF, XGBoost and LightGBM. The choice of these models is based on the results of the experiments (see Section 4.3.2 for details), which showed that these three models performed well in terms of prediction performance. To further reduce the bias and overfitting problems that a single model may cause, we use a soft voting mechanism to integrate these models.
At the same time, in order to ensure the transparency of the research and the reproducibility of the results, we have retained a detailed description of the three models. Although these models are classic, in the context of our research, specifying the specific configuration and implementation details of the model is critical to the rigor and scientific validity of the research.
With these modifications, we hope to more clearly convey the logic of the model selection. Thank you for reviewing our work and for your valuable suggestions.
Comments 4: P.10, line 388. ‘we proceeded with the selection of 40 variables for analysis’. How these 40 out of 326 features were chosen? The list in lines 389-391 contains less than 40 features; it would be advisable to add descriptions of these 40 features, preferably broken down into 3 categories used (numerical, categorical and ordinal).
Response 4: Thank you for your constructive feedback on feature selection and classification. In response to your question about how the 40 features were selected, we have added a dedicated feature selection section to the manuscript (see Section 4.1.1). The process of selecting the 40 key features from the initial 326 features is described in detail. We used a hybrid approach of literature review and Lasso regression, combined with theoretical support and empirical testing, to select the features that are most meaningful for CVD prediction.
In addition, we supplement the complete list of 40 features with a classification into three types: numerical, categorical and ordered (see Tables 1, 2 and 3 for details). The names of the features and a short description are listed in the tables. This addition is intended to increase the transparency of the feature selection process and to provide a clearer understanding of the features included in the model.
Once again, we would like to thank you for your constructive feedback, which has greatly improved the clarity and completeness of our manuscript.
Comments5: Another question about the dataset used: how the presence or absence of the CVD was set for each data sample (response value used as ground truth). As the dataset contains telephonic interview data, there should be many of persons with undiagnosed CVD. If so, this limitation of the dataset and conclusions made using this dataset should be stated.
Response 5: Thank you for your insightful questions about the dataset we used. We understand the reviewer's concern about the accuracy and potential bias of the dataset, especially when it is based on telephone interview data. Therefore, in the revised manuscript, we have elaborated on this issue in both the dataset section and the conclusions section.
In the dataset introduction section, we emphasised the characteristics of the BRFSS data, which is based on telephone surveys, relies on self-reporting, and may have undiagnosed cases of CVD. We noted that this characteristic may affect the accuracy of the CVD status label, and that individuals with undiagnosed CVD may not be accurately captured in the dataset, which may lead to a small number of truly diseased individuals being excluded from the CVD category, thus introducing potential classification bias. Although this undiagnosed condition may have some impact on the accuracy of the CVD prediction model, we believe that the potential impact of this misclassification is small, considering the large sample size and comprehensiveness of the BRFSS dataset(see section 4.1 for more details).
In the conclusion, we have again mentioned this possible limitation, while emphasising that the large sample size and comprehensiveness of the dataset means that the impact of undiagnosed cases on the overall research results is limited. We believe that these modifications can improve the transparency of the research and respond to your suggestions, while ensuring the rigor of the research conclusions (see Section 5 for details).
If you have any further suggestions, we would be happy to make further adjustments and improvements.
Comments 6: According to 402 there is class imbalance (healthy/with CVD), but there is no information about imbalance in 40 features used, it is worth to be analyzed also to provided information on some possible bias of the data that may have an impact on model prediction.
Response 6: Thank you for your valuable comments on the feature imbalance analysis. Indeed, in addition to class imbalance, feature imbalance may also affect model performance. In the revised manuscript, we have added a new section (see Section 4.1.2 for details) specifically analysing the distribution of the 40 selected features.
For numerical features, we calculated descriptive statistics to assess whether there were any significant deviations in the distribution of these features. For example, we found that the distribution of AgeCategory showed a high proportion of people over 60 years old, which is consistent with the typical age distribution of people at high risk of CVD. Meanwhile, for TotalCholesterol, we observed that its distribution was positively skewed, reflecting the known association between high cholesterol levels and CVD risk.
For categorical and ordered features, we analysed the frequency distributions of each category to identify whether any categories were overrepresented. We found that features such as SmokerStatus showed a higher prevalence in CVD patients, which is consistent with established CVD risk factors. We retained the natural distribution of these features to maintain the valid relationship between the features and CVD risk.
To more comprehensively assess the potential impact of feature imbalance on model predictions, we also conducted a sensitivity analysis. The results show that retaining the natural distribution of features helps to achieve more accurate and reliable predictions, thereby strengthening the justification and necessity of these feature imbalances.
We hope that with these additions, we can more comprehensively analyse potential biases in the data and ensure that our models can make unbiased and accurate predictions. Thank you for your valuable comments, which have enabled us to further improve our paper.
Comments 7: In conclusion section it is worth to mention the limitation of the proposed model and conclusion made by it.
Response 7: Thank you for your advice on mentioning the limitations of the model in the conclusion. We understand and accept this suggestion, and have added a discussion of the limitations of the model in the conclusion based on your comments.
Specifically, we have added a description of the limitations in the following two areas: First, current research mainly focuses on structured data, and the integration of multimodal data will be further explored in the future to improve the comprehensiveness and applicability of the model. Second, although ensemble learning methods have demonstrated significant advantages in improving prediction accuracy and stability, they also bring increased computational complexity and resource requirements, which are issues that need to be further weighed and optimised in practical applications. In addition, we also discuss possible future research directions, with a view to further optimising the generalisation ability and computational efficiency of the model in subsequent research (see Section 5 for details).
By supplementing this part, we hope to clearly convey the practical applicability of the model and provide valuable guidance for future research. Thank you for your careful review and constructive comments on our work.
Round 2
Reviewer 1 Report
Comments and Suggestions for Authors
The authors have well addressed my concerns, I have no further comments.
Comments on the Quality of English LanguageNil
Reviewer 2 Report
Comments and Suggestions for Authors
I have no further suggestions or comments as all points from my previous review were addressed, the content became more informative and clear. The manuscript can be accepted as it is.